# Intra-Species Genomic Variation in the Pine Pathogen *Fusarium circinatum*

**DOI:** 10.3390/jof8070657

**Published:** 2022-06-23

**Authors:** Mkhululi N. Maphosa, Emma T. Steenkamp, Aquillah M. Kanzi, Stephanie van Wyk, Lieschen De Vos, Quentin C. Santana, Tuan A. Duong, Brenda D. Wingfield

**Affiliations:** Department of Biochemistry, Genetics and Microbiology, Forestry and Agricultural Biotechnology Institute (FABI), University of Pretoria, Pretoria 0002, South Africa; mkhululi.maphosa@fabi.up.ac.za (M.N.M.); emma.steenkamp@fabi.up.ac.za (E.T.S.); aquillah.kanzi@fabi.up.ac.za (A.M.K.); stephanie.vanwyk@gmail.com (S.v.W.); santanaq@arc.agric.za (Q.C.S.); tuan.duong@fabi.up.ac.za (T.A.D.); brenda.wingfield@fabi.up.ac.za (B.D.W.)

**Keywords:** genome, accessory, core genome, *Fusarium circinatum*, structural variants, inversions, indels, pangenome

## Abstract

*Fusarium circinatum* is an important global pathogen of pine trees. Genome plasticity has been observed in different isolates of the fungus, but no genome comparisons are available. To address this gap, we sequenced and assembled to chromosome level five isolates of *F. circinatum*. These genomes were analysed together with previously published genomes of *F. circinatum* isolates, FSP34 and KS17. Multi-sample variant calling identified a total of 461,683 micro variants (SNPs and small indels) and a total of 1828 macro structural variants of which 1717 were copy number variants and 111 were inversions. The variant density was higher on the sub-telomeric regions of chromosomes. Variant annotation revealed that genes involved in transcription, transport, metabolism and transmembrane proteins were overrepresented in gene sets that were affected by high impact variants. A core genome representing genomic elements that were conserved in all the isolates and a non-redundant pangenome representing all genomic elements is presented. Whole genome alignments showed that an average of 93% of the genomic elements were present in all isolates. The results of this study reveal that some genomic elements are not conserved within the isolates and some variants are high impact. The described genome-scale variations will help to inform novel disease management strategies against the pathogen.

## 1. Introduction

Unravelling the genetic basis of population-level phenotypic variation, such as pathogenicity has been the underlying driving force in comparative genomics research [1,2]. For many fungal pathogens, we now know that genomes can be divided into sub-genomic compartments based on their evolutionary rates [3]. This is often referred to as the “two speed genome concept” [4], where some genomic regions seem to accumulate polymorphisms relatively quickly while others remain stable over extended periods of time. These more variable regions are often rich in repetitive elements and usually harbour genes that encode niche-defining phenotypes [5,6]. The latter includes effectors, which are secreted disease determinants of host infection and colonization, and are thus key to all plant–fungus interactions [7,8]. Variable regions are thus valuable resources for studying how fungal pathogens are adapted to their specific environments.

Although selection for disease phenotypes in plant breeding is dependent on knowledge about fungal pathogens’ inter- and intra-species variability [9,10], access to whole genome sequences can speed up the process. For example, virulence can be associated with specific micro and/or macro genomic polymorphisms that are located in specific sub-genomic compartments. The accuracy of such a process is therefore dependent on the quality of genome sequencing technologies, as well as genome assembly and variant calling platforms. Currently, long-read single-molecule sequencing technologies are useful for providing a broad scaffolding framework with which to assemble whole fungal genomes. This is because it can allow for the generation of long (up to 80 kb) sequence reads [11,12] that can span complex structural variants, including stretches of repetitive [12] and highly flexible regions which have shown to be associative with pathogenic determinants [1]. To account for the higher error rate of long-read sequencing technologies, they are often combined with short reads from platforms such as the Illumina technology [13]. Whole genomes in which chromosomes are sequenced from telomere to telomere can thus be assembled, thereby allowing a comprehensive analysis of genomic structural variation, as well as more fine-scale analyses, such as those involving single nucleotide polymorphisms (SNPs) [13].

In this study, we were interested in genomic variation in *Fusarium circinatum*. This economically important pathogen of pines is responsible for major losses in pine-based forestry around the world. Although high levels of genetic diversity and phenotypic variation have been reported within populations of this fungus [14,15], this was mostly attributed to sexual reproduction and mutation [16,17,18]. Other common sources of variation that could play a role in *F. circinatum* includes transposable elements [19,20] and horizontal gene transfer (HGT) [21]. Transposable elements can bring about genomic variations in several ways, e.g., transposition into exons, introns, regulatory regions and the mediation of non-homologous recombination [20]. To counter their impact on genome integrity, many fungi employ mechanisms such as repeat-induced point mutation (RIP) [22,23] that may introduce more variations, as have been shown in *F. circinatum* [24]. HGT is the exchange of genetic material between organisms that are not in a parent–offspring relationship [25]. A notable example in fungal pathogens are dispensable chromosomes, often harbouring pathogenicity genes, that can be exchanged among isolates [2]. In *F. circinatum*, the loss of a dispensable chromosome, in this case the 12th chromosome, reduces the pathogen’s virulence [26,27].

When trying to link a phenotype with genome-based polymorphism, a range of genomic features may be considered. These can range from micro variants, such as SNPs, to macro variants. The latter is mostly represented by structural variants (SVs) such as copy number variations (CNVs) or indels, translocations, inversions and whole chromosome loss/gain [28]. Although SNPs are most commonly employed, SVs also have great value as their occurrence and distribution can vary within populations [29] where they can have profound effects on phenotype diversity [30]. This is particularly true when SVs occur within genes or in regulatory regions [31,32]. CNVs or indels represent replicated/deleted genomic regions that arise from nonallelic homologous recombination and retrotransposition [33], and have shown to be capable of extensively impacting gene expression and phenotypic diversity [33,34]. Translocations result from recombination between nonhomologous chromosomes leading to reciprocal exchange or nonreciprocal transfer of the genetic material between the chromosomes [35]. Inversions are inverted chromosomal segments in which the DNA content remains the same and only the linear order of the DNA bases is changed [36]. They arise when a chromosome breaks at two points and the segment is reinserted in an inverted orientation. Inversions suppress recombination in heterokaryotypes and can even cause genetic isolation between populations [36,37]. However, relatively few studies have investigated SVs at the whole-genome level in fungi and/or attempted to associate them to phenotypic diversity [29,38,39].

The overall goal of the current study was to determine the extent to which SVs might influence the genome of *F. circinatum*. Our aims were four-fold: (i) sequencing of whole genomes for a set of geographically diverse *F. circinatum* isolates by making use of the short-read sequencing technology from Illumina^®^ (Illumina, San Diego, CA, USA), together with long-read sequencing technology from either PACBIO^®^ (Pacific Biosciences, Menlo Park, CA, USA) or MinION (Oxford Nanopore Technologies, Oxford Science Park, Oxford, UK); (ii) compilation of a comprehensive catalogue of fully characterized SVs occurring in the *F. circinatum* genome; (iii) annotation of genes associated with SVs; (iv) characterization of the *F. circinatum* pangenome for demarcating conserved and non-conserved genomic regions. This study will thus provide a valuable resource for future investigations on the functional effects of SVs on *F. circinatum* and contribute immensely towards the understanding of the biology of this pathogen, leading to the development of effective control mechanisms and management strategies.

## 2. Materials and Methods

### 2.1. Isolates

In this study we used seven isolates (FSP34, CMWF1803, CMWF560, CMWF567, UG10, UG27 and KS17) of *F. circinatum*. FSP34 was isolated from pitch canker-affected *Pinus* sp. in California, USA [38]. Isolates CMWF1803, CMWF560 and CMWF567 all originate from a pitch canker-affected *Pinus* species in Mexico [14]. Isolates UG10 and UG27 were isolated from pitch canker-affected *P. greggii* trees in a plantation near Ugie in the Eastern Cape Province of South Africa [17]. Isolate KS17 was isolated from the diseased roots of a *P. radiata* seedling that were obtained during a study of the pathogen in a commercial seedling production nursery in the Western Cape Province of South Africa [39].

### 2.2. Genome Sequencing, Data Sets and Assembly

High-quality DNAs were extracted from the respective fungi using lyophilised mycelia with the protocol described by Murray and Thompson [40]. For isolates CMWF560, CMWF567, CMWF1803, UG10 and UG27, Pacbio long reads (10 kb and 20 kb libraries) were generated by Macrogen (Seoul, South Korea), and for isolates KS17 and FSP34, MinION long reads were available in-house at the Forestry and Agriculture Biotechnology Institute (FABI, Pretoria, South Africa). For all the isolates, Illumina HiSeq2000 250bp paired-end sequencing was performed by Macrogen.

The Pacbio reads were filtered using the SMRT^®^ (Single Molecule, Real-Time) portal (Pacific Biosciences, Menlo Park, CA, USA), while MinION read correction and trimming was carried out using CANU [41]. The Illumina reads were trimmed and filtered using Trimmomatic v 0.38 [42], and FASTQC v 0.11.5 was used to check the quality of the trimmed reads. De novo genome assemblies were then generated with long reads using CANU [41]. The assemblies were initially polished with Quiver v 2.2.2 [43] and Nanopolish [44] using Pacbio and MinION raw reads, respectively. The final polishing was carried out with Pilon [45] using the Illumina HiSeq reads.

The polished scaffolds were ordered and oriented into contiguous pseudomolecules based on the macrosynteny that was found within the *Fusarium fujikuroi* species complex (FFSC) [46] using the LASTZ [47] plugin of Geneious v 7.0.4 [48]. The latter employed as references the chromosome-level genome assemblies of *F. fujikuroi* [49] and *F. temperatum* [50]. Following this, redundant contigs that showed high similarity with chromosome scaffolds or other longer contigs were discarded.

Quality checking on the final genome assemblies was carried out by mapping individual reads to the genomes of their respective genome assemblies and then performing a variant calling analysis. For this purpose, BWA MEM version 0.7.17-r1188 [51] with the -M option was used for the Illumina reads, and for the Pacbio and MinION reads we used CoNvex Gap-cost alignMents for Long Reads (NGMLR) version 0.2.7 [12] (accessed on 11 December 2021). After mapping, the SAM files were converted to BAM files using samtools view [52]. Read groups were then added using bamaddrg (https://github.com/ekg/bamaddrg, accessed on 11 December 2021), after which BAM files were sorted using bamtools sort [53]. Duplicated reads were marked with samtools rmdup. Samtools depth was used to determine the depth of coverage for each BAM file. The breadth of coverage was determined using samtools mpileup. For each genome, SVs were detected using Sniffles [12] with default settings.

Completeness of the various genomes was estimated using BUSCO (Benchmarking Universal Single-Copy Orthologs) v 2.0.1 [54] and the “Sordariomyceta” database containing 3725 genes. WebAUGUSTUS [55] was used to annotate the genomes with *F. graminearum* as the reference.

### 2.3. Identification and Annotation of SVs

To identify SVs, the genome assembly of isolate FSP34 was used as the reference. FSP34 was chosen for this purpose as it was the first *F. circinatum* isolate to have its genome sequenced [56], and has since undergone several re-sequencing and assembly improvements [57]. The version used here was further improved to an assembly of 12 chromosome-level scaffolds with 15 unmapped contigs. Quality-filtered reads from all sequencing platforms were used for mapping against the FSP34 genome assembly using BWA MEM for Illumina reads and NGMLR for Pacbio/MinION. As described above, samtools view was used to convert SAM files to BAM files, after which the read groups were added with bamaddrg. The BAM files were sorted with bamtools and duplicated reads were marked with samtools rmdup. SVs were then identified using Sniffles with default settings. To filter variants from low mapping quality regions we used samtools view to extract the low mapping quality reads (MQ < 5) from the sorted BAM files. Samtools depth was then used to compute the base coverage from the sorted BAM files with low mapping quality reads. We then used SURVIVOR bincov [28] to cluster the coverage track into a BED file for filtering. SURVIVOR filter was then used to filter the VCF files, which were sorted and merged using SURVIVOR. The merged VCF file was then used to force-call SVs across all samples. The resultant VCF files were merged to obtain a final multiple sample VCF file. The identified SVs were viewed with the Integrative Genome Viewer (IGV) v 2.4.14 [58].

To annotate the identified SVs with SnpEff [59], we first compiled a SnpEff database for *F. circinatum*. This was completed using the annotation files that were obtained from WebAUGUSTUS. After the SVs in the VCF files were annotated, we used SnpSift [60] to extract genes with high or moderate impact variants and genes from chromosomal regions with high variant density for further analysis. These genes were subjected to a gene ontology (GO) enrichment analysis using the Fisher test (*p* value < 0.05) in the Blast2GO [61] plugin in CLC Genomic Workbench (Aarhus, Denmark). The list of all predicted genes from the FSP34 reference genome were used as the reference set for this analysis.

All reads that did not map to the FSP34 assembly were filtered from the respective BWA-MEM and NGMLR alignments. These were then assembled using SPAdes version 3.7.1 [62]. The assembled contigs were annotated using WebAUGUSTUS as described above. A BLASTn and BLASTp analysis of these predicted annotations was conducted using NCBI.

### 2.4. Identification and Characterization of Micro Variants

For the micro variant analysis, we specifically targeted SNPs, multiple nucleotide polymorphisms (MNPs) and indels in the 1–29 base pair (bp) range. For this purpose, the BAM files generated above were subjected to analysis with Freebayes version v1.2.0-2-g29c4002 (accessed on 11 December 2021) [63] using the options “—ploidy 1”; “—min-mapping-quality 30”; “—min-base-quality 20” and default settings for the rest of the parameters. Filtering of the generated variants was carried out using vcffilter (https://biopet.github.io/vcffilter/0.2, accessed on 11 December 2021). Variants with quality scores (QUAL) that were greater than 30 and with a minimum depth of coverage (DP) greater than 10 were filtered and removed. The identified micro-variants were then annotated using SnpEff, SnpSift and Blast2GO as described above.

### 2.5. Synteny and Pangenome Analyses

Whole genome alignments were conducted using LASTZ as described above. Pairwise alignments between the genomes were visualized in Geneious. A synteny analysis was carried out using Synchro [64] with the protein FASTA files that were derived from the WebAUGUSTUS annotations. Visualization of SVs was also carried out as graphic outputs from Synchro.

Spine [65] was used to reconstruct the pangenome of *F. circinatum* from the seven genomes that were examined. This program uses NUCmer [66,67] to align whole genomic sequences to build a nonredundant pangenome and extracts the core and accessory genomic elements as defined in the input parameters [65]. The program can take annotated genomes and output protein information online, but this function currently only works for small data sets, such as prokaryotic genomes. For our purposes, we downloaded the program and ran the analysis on our local servers. For this study, we used whole genome FASTA files and then searched for genes in the resultant data sets using WebAUGUSTUS, as described above. Similarly, BUSCO analysis was carried out on all predicted proteins.

Genomic sequences that were present in all seven genomes, with >85% similarity and a minimum output fragment of 100 bp and a gap of 10 bp between fragments, were regarded as belonging to the core sub-genomic compartment. To determine the distribution of accessory genomic elements among isolates we used the ClustAGE [68] package in Spine. From the ClustAGE bins we selected bins that had sequences >4.5 kb for further analysis of their distribution patterns within the respective genomes. We focused on two groups of accessory genomic elements, the uniquely absent (sequences that were present in 6 isolates and missing only in 1 isolate) and the uniquely present (sequences that were present only in a single isolate). We used BLASTp analysis in NCBI to determine the origins or possible sources of these sequences. Geneious was used to map the accessory genomic elements to the genomes to check their distribution across the twelve chromosomes.

## 3. Results

### 3.1. Genome Sequencing, Data Sets and Assembly

We used long reads and short reads to construct five new and nearly complete chromosome-scale genome assemblies for *F. circinatum* (Table 1). Long-read sequences that were generated with either PacBio or MinION yielded a total of 1,557,037 filtered, corrected and trimmed reads for the seven isolates. Illumina sequencing generated a total of 64,129,476 reads after filtering and trimming. Most of the Illumina reads had a base quality sequence score that was greater than 30. Together with the two previously sequenced isolates (FSP34 and KS17), the total genome sizes ranged from 45,008,552 bp to 43,828,286 bp (Table 2). All genomes were assembled to the expected 12 linear chromosome scaffolds and the single circular mitochondrion scaffold. BUSCO analyses indicated that genome completeness ranged from 98.2% to 99.1% (Table 1). The number of complete genes that was predicted for these genomes by WebAUGUSTUS ranged from 13,854 to 14,382.

The results of our read mapping experiments suggested that the generated assemblies were robust. Mapping reads to their corresponding genome assemblies revealed that sequencing depth for long-read PACBIO and MinION sequencing ranged from 14X to 74X coverage, while for short-read Illumina sequencing ranged from 40X to 60X coverage. Further, no variants were detected on all of the seven genome assemblies when SV calling was carried out with reads that were mapped to their corresponding assemblies. It is thus unlikely that the genome assemblies have errors that could result in false calls of variants based on the reference genome and whole genome comparisons.

For all isolates, sequencing data that could not be assembled to the 12 chromosome scaffolds and mitochondrion ranged from 114,529 bp for isolate FSP34 to 1,142,366 bp for isolate UG10. The latter corresponded to 42 contigs ranging in size from 1992 bp to 65,074 bp. Overall, however, the unmapped contigs from all assemblies showed high similarity with fragments from the 12 core chromosomes (Figure 1), but due to their smaller size we could not accurately position them in their respective chromosomes.

Additionally, a scaffold that could not be assembled to any of the 12 chromosomes and that was larger in size (1,045,806 bp) than chromosome 12 was observed for isolate CMWF1803. Comparisons with the other isolates revealed a similar scaffold with high similarity in isolate CMWF560 (Figure 1). These two isolates also had a higher overall genome size compared to the rest of the isolates. This uncharacterized scaffold might be an additional chromosome for *F. circinatum*.

### 3.2. Identification and Annotation of SVs

Of the short reads that were generated for the different genomes, 90.41% to 99.0% mapped to the FSP34 assembly, while 93.7% to 99.2% of the long reads mapped to it (Table 3). Following the construction of BAM files and multi-sample variant calling with Sniffles, we identified a total of 1 828 SVs, ranging in size from 30 bp to >10,000 bp (Table 4: Appendix A). Of these, 1717 were copy number variants (990 deletions, 719 insertions, 8 duplications) and 111 were inversions.

Annotation with SnpEff predicted a total of 476,229 functional effects for the collection of SVs that were identified (Table 5: Appendix A). SnpEff also provided an indication of the magnitude of the predicted effects. Variants with a high impact are disruptive with regards to probable protein function; moderate impact variants are less disruptive and would likely reduce functionality of the protein; low impact variants would be mostly harmless; and modifier variants often occur in non-coding genes, introns, intergenic regions, intragenic regions and downstream to genes where predictions of the effect are more difficult [59]. For the SVs identified here, a greater proportion (88.2%) were predicted to have a high impact, 10.5% a moderate impact, 0.0% (a single effect count) a low impact and 1.3% a modifier impact. The high impact-effect variant types included bidirectional gene fusions (46.6%); chromosome number variations (0.002%); exon loss variants (0.02%); feature ablations (1.6%); frame shift variants (0.03%); gene fusions (37.9%); inversions (10.5%); splice acceptor variants (0.003%); splice donor variants (0.004%); start losses (0.01%); stop losses (0.008%); and transcript ablations (2.0%). Moderate impact-SV variant types included conservative in-frame deletions (0.02%); disruptive in-frame deletions (0.01%); and upstream gene variants (0.5%). Modifier impact variant types that were predicted included downstream gene variants (0.5%); exon regions (0.0%); intergenic regions (0.13%); intergenic variants (0.08%); intron variants (0.01%); non-coding transcripts variants (0.07%); and splice region variants (0.01%).

A total of 4190 proteins were annotated in the regions that were predicted to have deletions, 345 in regions displaying insertions, 8885 were in inversion areas and one protein was in an area displaying duplications (Appendix A). The following biological processes were overrepresented in the gene set of proteins that were annotated in regions associated with deletions (Figure 2): DNA metabolism, cellular protein modification process, chromosome organization, response to stress, signal transduction, macromolecular complex assembly, tRNA metabolism, chromosome segregation, cell division, mitotic cell cycle, small molecule metabolism, biosynthesis, cofactor metabolism, reproduction, cell adhesion and intracellular transport. The overrepresented cellular components with the deletions were macromolecular complex, nucleoplasm, chromosome, ribosome, endosome, cytosol, cytoskeleton, mitochondrion and nuclear envelope. Molecular functions that were overrepresented within proteins displaying deletions were enzyme regulator activity, kinase activity, nucleotidyltransferase activity, cytoskeleton protein binding, enzyme binding, protein binding, ion binding, helicase activity, rRNA binding, ligase activity, isomerase activity, rRNA binding and translation factor binding.

Three biological processes were overrepresented in the predicted gene set that was predicted to be affected by insertions (Figure 3). These were cell adhesion, nickel cation transmembrane transport and ubiquitin-dependent protein catabolism. Cellular components that were overrepresented in insertions were integral components of the membrane, the proteasome core complex and the alpha subunit. Dominant molecular functions in this gene set were threonine-type endopeptidase activity, amidase activity, nickel cation transmembrane and transporter activity.

Six biological processes and four cellular components were overrepresented in the genes that were affected by inversions (Figure 4). The biological processes were cofactor metabolism, protein targeting, homeostatic process, carbohydrate metabolism, cell division and response to the stimulus. Overrepresented molecular functions were transferase activity, transferring phosphorus-containing groups, RNA binding, protein binding and enzyme regulator activity.

The single gene that was predicted to be affected by duplications encoded an RNA binding protein with helicase activity, which is a ubiquitous family of proteins that potentially play a role in cellular processes that involve RNA metabolism [69]. The unmapped reads represented loci that were absent in the reference strain and/or contaminants in our samples. The BLASTp analysis retained top hits from the *Fusarium* species. Genes that were annotated from these sequences were mainly for proteins that were involved in transport and metabolism (Figure 5).

### 3.3. Identification and Annotation of Micro Variants

Multi-sample variant calling using Freebayes generated a total of 397,704 polymorphic sites among the genomes that were examined (Appendix A). Of these, 353,117 (88.8%) were biallelic and 44,587 (11.2%) were multiallelic. From the collection of polymorphisms, a total of 461,683 micro variants were identified, of which 68.7% were SNPs, 23.1% were 1–29 MNPs, 3.2% were insertions, 2.4% were deletions and 2.7% were mixed. The observed genome-wide transitions/transversions (Ts/Tv) ratio was 2.9 and the overall variant rate was one variant for every 97 bases. The observed Ts/Tv ratio is consistent with the phenomenon that transitions generally occur at a higher frequency than transversions [70]. The whole genome SNP density showed an uneven distribution of SNPs across chromosome lengths. There were high SNP densities on chromosome arms while the middle regions showed relatively lower SNP densities (Figure 6). For chromosome 11 and 12, the SNP densities were relatively high across full chromosome lengths.

A total of 2,719,213 putative effects were assigned to a collection of micro variants. Of these, SnpEff predicted 0.7% of the effects to have a high impact, 2.3% to have a moderate impact, 2.8% to have a low impact and 94.3% to have a modifier effect. In terms of functional classes, 42.0% of the effects were classified as missense, 0.9% nonsense and 57.1% to be silent variants (Table 6 and Appendix A). High impact variants included frameshifts, gene fusions, splice acceptors, splice donors, start losses, stop gains and stop losses. The predicted moderate impact variants were conservative in-frame deletions, conservative in-frame insertions, disruptive in-frame deletions, disruptive in-frame insertions, missense variants and upstream gene variants. Low impact variants included initiator codon variants, splice region variants, stop retained variants and synonymous variants. Modifier variants included were downstream gene variants, intergenic region, intragenic variants, intron variants and non-coding transcript variants.

The bulk of the predicted effects were outside the coding regions, in downstream, intergenic, intron and upstream regions. A total of 136,959 effects were predicted to impact on exons, 13,881 on genes and 30,148 on coding and non-coding transcripts. Within the splice sites, 266 effects were predicted on the splice site acceptor region, 301 on the splice site donor and 4518 within the splice site region.

A GO enrichment analysis showed that genes that were involved in the following biological processes were overrepresented (*p* > 0.05) in chromosome regions with high micro variant densities (Figure 7): transmembrane transport, carbohydrate metabolism, phosphorylation, proteolysis, eisosome assembly, secondary metabolism, oxidation-reduction process, nucleoside metabolism, transcription, DNA-templated and regulation of nitrogen compound metabolism. These overrepresented genes are localised within the integral component of membrane, the extracellular space and the eisosome. Overrepresented molecular functions within this gene set were oxidoreductase activity, transcription factor activity, sequence-specific DNA binding, hydrolase activity, acting on glycosyl bonds, hydrolase activity, activity on carbon-nitrogen (but not peptide) bonds, peptidase activity, acting on L-amino acid peptides, zinc ion binding, ADP binding and transmembrane transporter.

Genes that were overrepresented in gene sets affected by high impact micro variants included the following biological processes: transmembrane transport, regulation of transcription from RNA polymerase II promotor, RNA phosphodiester bond hydrolysis, eisosome assembly, cell adhesion and phosphorylation (Figure 8). The overrepresented cellular components were an integral part of the membrane and eisosome. Overrepresented molecular functions included oxidoreductase activity, zinc ion binding, calcium ion binding, ATP binding, RNA polymerase II transcription factor activity, sequence-specific DNA binding, transmembrane transporter activity and ribonuclease activity.

Protein classes that were overrepresented in regions affected by moderate impact micro variants (Figure 9) included these biological processes: transmembrane transport, carbohydrate metabolism, regulation of transcription from RNA polymerase II promoter and phosphorylation. Molecular functions that were overrepresented in genes affected by moderate variants were: oxidoreductase activity, hydrolase activity, acting on glycosyl bonds, DNA binding, hydrolase activity, acting on carbo-nitrogen (but not peptide) bonds, zinc ion binding, ATP binding, RNA polymerase II transcription factor activity and sequence-specific DNA binding.

### 3.4. Synteny and Pangenome Analyses

Using Spine, a nonredundant pangenome of 50,076,541 bp was built from the seven genomes that were used in this study. WebAUGUSTUS predicted a total of 15,099 complete genes from the pangenome (the pangenome was 96.7% complete according to BUSCO with 76 fragmented BUSCOs and 47 missing BUSCO genes). The percentage completeness of the pangenome is lower than the individual genomes of the seven isolates, which is indicative of the fragmentation of the pangenome. Analyses of the combined proteins that were annotated from the seven isolates indicated that there were 18 missing BUSCO genes based on the “Sordariomyceta” database. BLASTp analyses of the 18 missing BUSCOs revealed that only four of the proteins had top hits to *Fusarium* species, suggesting that these missing genes may not be common among *Fusarium* species. The total combined number of annotated proteins from the seven genomes was 98,484, of which 99.2% were clustered into orthogroups [72] with an average of 7.1 genes per orthogroup. Sequences from all isolates were present in 12,424 orthogroups, of which 11,712 represented single copy gene families. A combined total of 765 genes were not assigned to any orthogroup. These were mainly the isolate-specific genes based on the sample that was used in this study.

Whole genome alignments using Spine resulted in a backbone (core) of 42,260,189 bp that was present with a nucleotide similarity greater than 85% in all seven of the genomes (Table 7). This core comprised an average of 93% of all the seven genomes used in this study and had a GC content of 47.2%. A total of 13 282 complete genes were predicted by WebAUGUSTUS in the core genome. A comparison against BUSCO’s “Sordariomyceta” databased indicated that the core genome was 95.6% complete. The BLASTp analysis of the core proteins, excluding hits on *F. circinatum,* showed 13,182 proteins had top hits to *Fusarium* species, 48 proteins had top hits to non-*Fusarium* species and 52 predicted proteins had no blast hits to the NCBI database (Appendix A). Non-*Fusarium* top hits included species from the genera *Neonectria*, *Trichoderma*, *Exophlala* and *Aspergillus*. Biological processes that were overrepresented within the set of 48 non-*Fusarium* proteins were regulation of transcription by RNA polymerase II, methylation and phosphorylation. Overrepresented molecular functions within the 48 proteins were DNA-binding transcription factor activity, RNA polymerase II specific and zinc ion binding.

An average of 7% of the genome sequence across the seven isolates was predicted to constitute the accessory genomic elements (Appendix A). These were distributed across chromosome lengths, with chromosomes 1 to 11 showing higher densities in the sub-telomeric regions, while chromosome 12 and the potential additional chromosome (UC > 500 Kb) had relatively higher densities across full chromosome lengths (Appendix A). The number of complete genes in the accessory genomic elements of the isolates was 58,881, of which 64 were complete BUSCOs. The combined accessory genomic elements from all the isolates amounted to a total of 22,175,551 bp with an average GC content of 44.8%. ClustAGE grouped sequences that were greater than 200 bp into 2023 unique bins of similar sequences.

A total of 632 genes were annotated on sequences that were only present in one of the examined genomes. Of these, 596 had best BLAST hits in other *Fusarium* species, 15 had similarity in non-*Fusarium* species and 20 genes returned no significant result using BLAST. Isolate CMWF1803 had the highest number of such uniquely present genes (240) while none were found in isolate UG10. Isolates CMWF560, CMWF567, FSP34, KS17 and UG27 had 148, 143, 57, 40 and 3 uniquely present genes, respectively. From the sequences that were absent in only one of the genomes examined, a total of 466 genes were annotated. Of these, 445 had best BLAST hits to other *Fusarium* species and 21 had best BLAST hits to non-*Fusarium* species. Isolate FSP34 had the highest number of uniquely absent genes (150) followed by KS17 (140) while none were found in isolate UG27. Isolates CMWF560, CMWF567, CMWF1803 and UG10 had 72, 47, 21 and 36 uniquely absent genes, respectively.

A total of 18 biological processes were overrepresented in the uniquely present proteins. These included tRNA metabolism, chromosome segregation, cellular protein modifications, ribonucleoprotein complex biogenesis, anatomical structure formation involved in morphogenesis, nucleocytoplasmic transport, chromosome organization, cell differentiation, nucleocytoplasmic transport, translation, DNA metabolism, protein transport, sulphur compound metabolism, response to stress, cofactor metabolism and cellular catabolism. Dominant molecular functions in this set were cytoskeletal protein binding, structural constituent of ribosome, RNA binding, phosphatase activity, nuclease activity, GTPase activity, protein binding, bridging and ubiquitin-like protein binding. Overrepresented cellular components in this set were macromolecular complex, cytoskeleton, nuclear chromosome, mitochondrion, Golgi apparatus, ribosome, nuclear envelope and extracellular matrix.

Among the uniquely absent proteins, the following biological processes were overrepresented: mitotic cell cycle, tRNA metabolism, reproduction, anatomical structure morphogenesis, chromosome segregation, cellular protein modification process and organelle organization. Molecular functions that were overrepresented in this set included peptidase activity, nuclease activity and phosphatase activity. The dominant cellular components were macromolecular complex, nuclear chromosome, extracellular matrix and endomembrane system.

## 4. Discussion

In this study, we used multi-sample variant calling to develop a catalogue of intra-species SVs for the pine pitch canker fungus, *F. circinatum*. This type of resource is relatively uncommon for fungi, as most previous studies explored genomic variation at the inter-species level [2,5]. These precious comparative genomics studies have provided insights into the biology of closely related species and improved our understanding of how they evolve [5,73]. By contrast, studies on the intra-species genetic variation provide insights into how individuals of the same species accumulate genetic differences and how these affect their adaptability [14,74,75]. As genomes of many isolates of the same species become available it will become routine for whole genome comparative studies to be conducted and enable phenotypic traits to be associated with genotypic variations.

To facilitate the compilation of our catalogue of SVs, we first had to ensure that we had access to high-quality reference assemblies. Most variant calling platforms use a reference genome to identify polymorphisms in other genomes of interest [12,63]. However, considering the broad structural variation that exists within fungal genomes [76], reference genomes can be limiting in variant calling if the assembly is incomplete or if it has assembly errors, which can lead to mapping errors that distort variant calling [77]. Indeed, it is widely recognized that the availability of good reference genomes against which comparisons can be made represent one of the main limitations of efficient and accurate in variant calling [78]. The current genome assembly for *F. circinatum* isolate FSP34 is near-complete, containing 12 chromosome scaffolds and the mitochondrion scaffold [57]. This isolate has also been used in various previous studies [46,74,75,79] and has emerged as the “model” strain for *F. circinatum* genomics studies.

In addition to the FSP34 reference genome, we also report here the high-quality genome assemblies for five additional *F. circinatum* strains. They have different origins, with three (CMWF560, CMWF567 and CMWF1803) originating from pitch canker-affected *Pinus* spp. tissue that was collected in Mexico [14] and two (UG10 and UG27) that were isolated from pitch canker-affected *P. greggii* tissue in South Africa [17]. The strains also have different mating types. *Fusarium circinatum* is heterothallic [80], with three of the isolates (CMWF560, CMWF567 and CMWF1803) being MAT 1–1 and two (UG10 and UG27) MAT 1–2. The five additional genomes were also assembled to near completion, with all 12 of the expected chromosome scaffolds. The 12th chromosome is the smallest and apparently dispensable [2,26,81].

In two of our new genome assemblies we detected a scaffold that likely represents an additional chromosome. These two chromosome-sized scaffolds were found among the uncharacterized scaffolds in the CMWF560 and CMWF1803 sequence data. Subsequent analyses confirmed that they were found only in isolates CMWF560 and CMWF1803, but not in the other five *F. circinatum* isolates, including FSP34. Chromosome number variation has been reported within species of the FFSC, where at least two dispensable or accessory chromosomes have been visualised [82]. We thus propose that the uncharacterised scaffolds identified here are an additional accessory chromosome, which we call chromosome 13. The 13th chromosome in CMWF1803 was the larger of the two at 1 045 806 bp, which is bigger than chromosome 12 assembled across all the isolates. The overall genome sizes of isolates CMWF560 and CMWF1803 were generally bigger than the rest of the isolates that did not have sequences resembling the additional chromosome.

Our results showed that micro variants were unevenly distributed across the lengths of chromosomes 1–10. In these chromosomes, we observed higher densities over the first 1Mb of chromosome ends, while they were more sparsely distributed in the middle regions of the chromosomes. This distribution pattern is consistent with previous reports of high variant density on sub-telomeric regions [31,32], which are widely recognized as hot spots for genome plasticity in eukaryotes. This is because telomeres accumulate variants during replication as they replicate differently from the inner regions of chromosomes [83]. Further, sub-telomeric regions are prone to recombination and are thus likely to accumulate mutations [84]. In contrast to chromosomes 1–10, chromosomes 11 and 12 generally had high micro variant densities across full chromosome lengths. The sequence and size variability of chromosome 12 [57,85], and the presence of a large reciprocal translocation and an inversion in chromosome 11 [46,50] make chromosomes 11 and 12 the most variable of the chromosomes within the FFSC [50]. This is evidenced in our results as the higher variant density across the full chromosome lengths.

Copy number variants (insertions, deletions and duplications) ranging from SVs as small as 30bps to large ones of >100 kbs were detected in this study. These types of variants are known to play a major role in fungal adaptation [29,86]. Duplications can influence the expression levels of target regions, e.g., duplication of regions containing biosynthetic genes can increase production of the relevant compounds [81,87,88], or duplication of certain genes can increase resistance to toxic compounds [89]. In this study, we identified a gene encoding an RNA binding protein with helicase activity arising by duplication. Furthermore, insertions can drive the accumulation of novel genetic traits, while deletions highlight genomic regions that might be dispensable as pathogenic fungi often shed genomic regions to avoid detection by hosts [6]. Therefore, the copy number variants that were identified in this study provide a robust starting point for further research into the biology of the pitch canker fungus.

Structural variants that impact genes can lead to functional variation among individuals and influence phenotypic traits. We used SnpEff [59] to access the significance of the identified variants based on gene models from the WebAUGUSTUS annotations [55]. Although the accuracy of our gene models still needs to be verified, the identified putative effects provide a basis for future association studies using these isolates. The most prevalent high impact variant effect was gene fusion. Clustering of genes involved in the same metabolic pathway, biological process or structural complex is a well-known phenomenon in fungi [90]. This has a beneficial effect because it allows for the physical coupling of proteins that are functionally related. Inversion was also a common high impact macro structural variant effect. Inversions are indicative of genes whose transcriptional orientation along chromosomes is different [91]. Frameshift mutations are indels within the coding regions of genes that alter the reading frame, resulting in alteration of the downstream codons. In many cases, the deletion or insertion results in truncated gene products [92]. Splice site mutations would result in abnormal splicing leading to base changes in the processed mRNA [93,94]. The phenotypic effect of all these mutations on the different isolates of *F. circinatum* needs to be investigated. GO enrichment analyses showed that genes that were involved in transcription, DNA binding, transmembrane transport, oxidoreductase activity, hydrolase activity, acting on glycosyl bonds, ion binding, metal binding and cation binding were overrepresented in regions that were affected by high and moderate impact variants. These classes of genes are mainly involved with proteins that are needed by the fungus when interacting with its environment. The presence of variants in these gene classes highlights the pathogen’s propensity to adapt to its environment.

We took a sequence-centric approach [95] in building the pangenome of *F.*
*circinatum*. We characterized the core, accessory and pangenomes based on nucleotide sequence rather than only the protein-coding portion of the genomes. This approach ensured the inclusion of noncoding genomic elements in the core, accessory and pangenomes. This methodology allowed us to avoid the bias and errors of gene prediction models in the identification of genomic elements that are conserved in *F. circinatum*. Our BLAST analysis showed that within the core genome there were some proteins which did not return significant hits to closely related *Fusarium* species, but rather to other, distantly related species. These genomic elements may have been acquired by *F. circinatum* from other species through horizontal gene transfer [96] and have been subsequently maintained, possibly as they have provided some advantage to the species. Gene gains can also be attributable to de novo gene emergence from non-coding DNA. There were also some proteins that did not return any BLAST hits. These were mainly proteins of unknown function.

These orphan genes which lack homologues in other lineages can also arise through duplications followed by diversification [96]. The proteins that are unique to *F. circinatum* and are conserved in all the isolates could be involved in lineage-specific adaptations and are potential targets for species-specific pathogen management and control strategies.

Within the accessory genome, isolate-specific open reading frames that were only present in one isolate and not the others (uniquely present) also showed different patterns of possible origins. Predicted proteins that had BLAST hits with closely related *Fusarium* species could indicate genomic elements that were maintained in some isolates but dispensed in others. These genomic regions could have arisen through high impact variants which include deletions, loss of function mutations (such as frame shifts) and early stop codons, resulting in truncated proteins. Even if truncated proteins are expressed in different isolates, they are less likely to have exactly the same function as full-length proteins. Therefore, the functional variation of truncated proteins could have phenotypic effects on different isolates. Isolate-specific sequences with best BLAST hits to other non-*Fusarium* species could be indicative of recently gained genomic elements that have not been fixed in *F. circinatum* populations. The accessory genome elements also had a reduced GC content, in comparison to the core genome. There were more accessory elements mapping to chromosomal regions that also had a high SNP density. Accessory genomic elements were associated with AT rich and highly variable genomic regions.

## 5. Conclusions

The genomic structural variants that were identified in *F. circinatum* ranged from SNPs to chromosome scale SVs, including insertions, deletions and inversions. It is possible that some of these variations are neutral with little or no impact on the biology of this pine pathogen. However, it is also conceivable some of the SVs that are described in this study have profound effects on the biology, evolution and niche adaptation of this pine pathogen. The genetic variations could then impact the disease management and control strategies that are used in nurseries and plantations, as these strategies need to be applicable to strains of the pathogen notwithstanding variations between the strains. Therefore, this study provides a resource for future association studies within *F. circinatum*. Progress in understanding the exact impact of these genetic variations within *F. circinatum* will also benefit our understanding of the biology of the pitch canker fungus.

## Figures and Tables

**Figure 1 jof-08-00657-f001:**
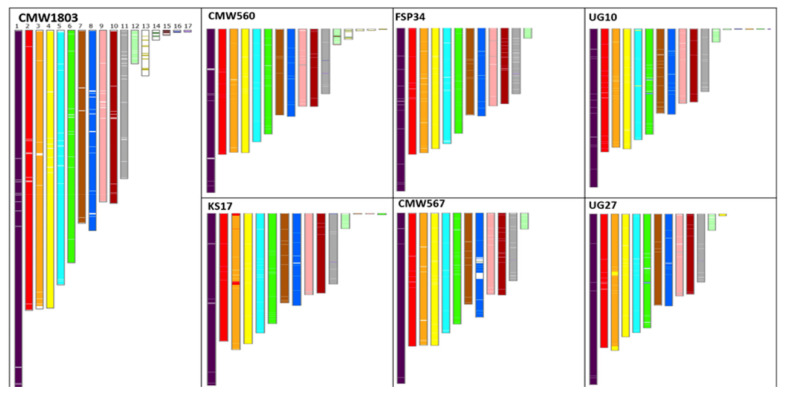
SynChro [64] generated chromosome painting, comparison of the isolates showing the uncharacterized scaffold > 500 Kbs in CMWF1803 and CMWF560. No similar sequences were found in the other isolates.

**Figure 2 jof-08-00657-f002:**
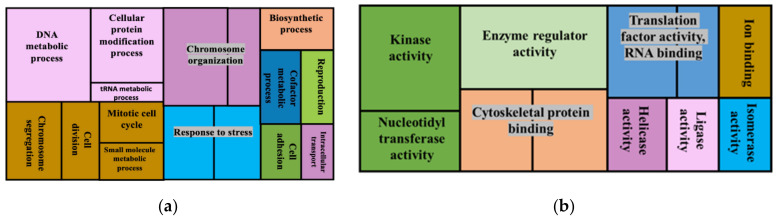
GO enrichment analysis for genes associated with deletions showing overrepresented GO terms, biological processes (**a**) and molecular functions (**b**). The whole genome annotations of the reference FSP34 were used as the reference set and genes associated with deletions were used as the test set.

**Figure 3 jof-08-00657-f003:**
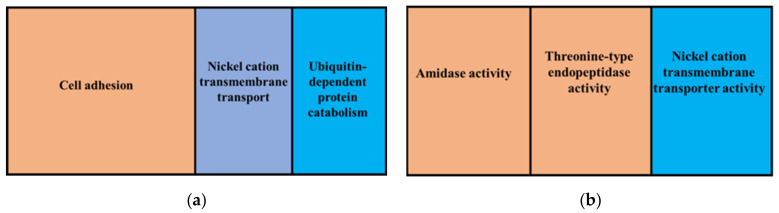
GO enrichment analysis for genes affected by insertions showing overrepresented GO terms, biological processes (**a**) and molecular functions (**b**). The whole genome annotations of the reference FSP34 were used as the reference set and genes affected by insertions were used as the test set.

**Figure 4 jof-08-00657-f004:**
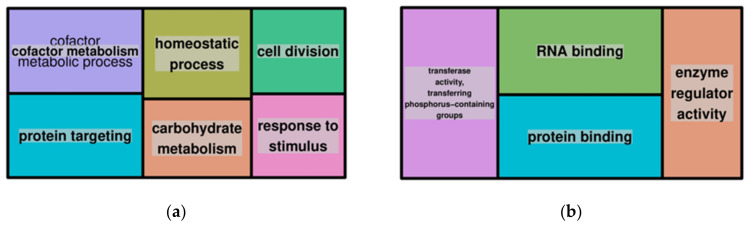
GO enrichment analysis for genes associated with inversions showing overrepresented GO terms, biological processes (**a**) and molecular functions (**b**). The whole genome annotations of the reference FSP34 were used as the reference set and genes within inverted regions were used as the test set.

**Figure 5 jof-08-00657-f005:**
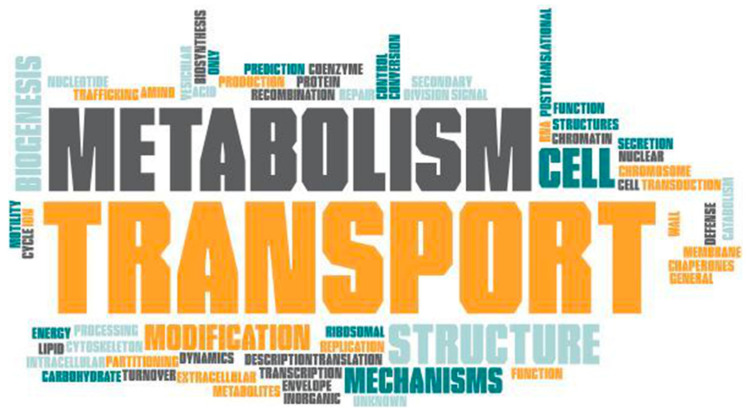
Protein classes coded by genes found in assembled sequences that did not map to the FSP34 genome assembly.

**Figure 6 jof-08-00657-f006:**
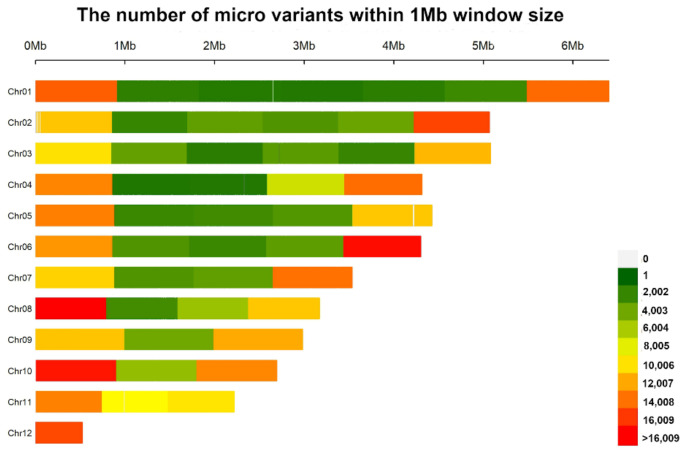
Micro variant density distribution across chromosome lengths of *F. circinatum* FSP34 relative to the other 6 genomes showing regions of higher variant density in chromosome arms and sparse distribution within the inner central parts of the chromosomes. The figure was generated using CM plot [71].

**Figure 7 jof-08-00657-f007:**
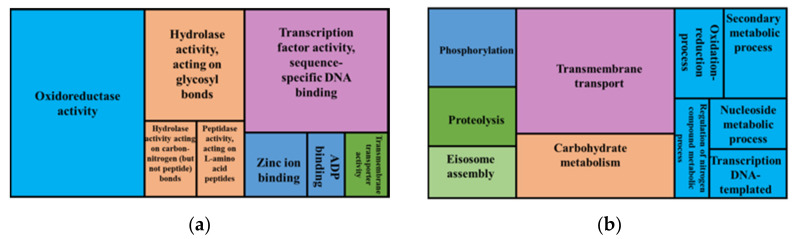
GO enrichment analysis for genes within high variant density chromosomal regions showing overrepresented GO terms, biological processes (**a**) and molecular functions (**b**). The whole genome annotations of the reference FSP34 were used as the reference set, genes within high variant density regions were used as the test set.

**Figure 8 jof-08-00657-f008:**
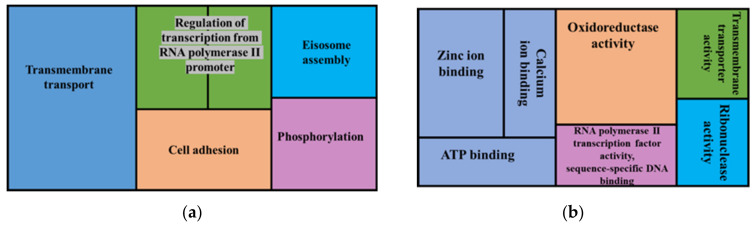
GO enrichment analysis of genes predicted to have high impact variants showing overrepresented GO terms, biological processes (**a**) and molecular functions (**b**). The whole genome annotations of the reference FSP34 were used as the reference set and genes predicted to have high impact variants were used as the test set.

**Figure 9 jof-08-00657-f009:**
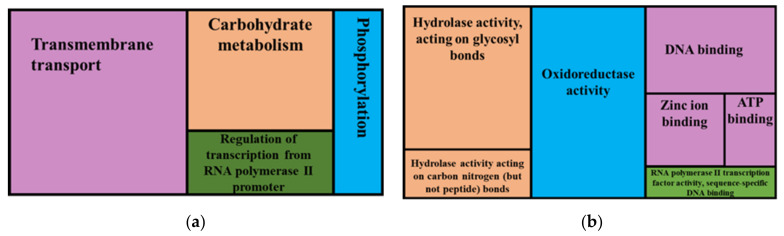
GO enrichment analysis for genes predicted to have moderate impact variants showing overrepresented GO terms, biological processes (**a**) and molecular functions (**b**). The whole genome annotations of the reference FSP34 were used as the test set and the genes with moderate impact variants were the test set.

**Table 1 jof-08-00657-t001:** Summary of whole nuclear genome assembly statistics.

Isolate	CMWF560	CMWF567	CMW1803	UG10	UG27
Accession number	JAEHFI 000000000	JADZLS 000000000	JAEHFH 000000000	JAGJRQ 000000000	JAELVK 000000000
Genome size (bp)	46,691,343	45,984,420	46,810,763	44,774,968	45,546,500
Genome coverage ^1^	35 (58)	65 (56)	54 (56)	31 (60)	43 (43)
N50	4,436,154 bp	4,431,017 bp	4,492,802 bp	4,380,615 bp	4,358,900 bp
N75	3,211,240 bp	3,566,220 bp	3,263,251 bp	3,014,022 bp	3,202,209 bp
L50	5	5	5	5	5
L75	8	8	8	8	8
% G+C	46.78	46.87	47.05	47.50	46.83
BUSCO (%)	99.0	99.0	99.1	98.9	99.1
Number of chromosomes	12	12	12	12	12
Uncharacterised contigs	37	12	7	16	35
ORF	14,170	14,116	14,382	14,094	13,987

^1^ Genome coverage for Pacbio sequencing, with Illumina sequencing indicated in brackets ().

**Table 2 jof-08-00657-t002:** Size of the different scaffolds in each of the assemblies for the seven *Fusarium circinatum* isolates used in this study.

Scaffold ^1^	*F. circinatum* Isolate ^2^						
	FSP34	CMWF560	CMWF567	CMWF1803	UG10	UG27	KS17
Chr01	6,407,689	6,519,590	6,423,820	6,503,389	6,400,822	6,430,758	6,397,914
Chr02	5,066,197	5,005,606	5,040,005	5,011,558	4,859,649	4,976,076	4,709,326
Chr03	5,081,888	5,037,826	5,141,289	5,085,556	4,913,444	5,257,461	5,148,568
Chr04	4,313,168	4,556,945	4,479,672	4,551,627	4,383,147	4,236,059	4,401,926
Chr05	4,432,553	4,436,323	4,431,017	4,492,934	4,408,860	4,425,559	4,304,443
Chr06	4,301,895	4,284,508	4,281,318	4,282,150	4,261,236	4,358,679	4,219,930
Chr07	3,541,054	3,578,508	3,565,977	3,555,373	3,412,693	3,591,987	3,312,103
Chr08	3,172,915	3,210,569	3,718,348	3,263,924	3,015,014	3,202,361	3,066,990
Chr09	2,981,544	2,920,641	2,844,992	2,857,338	2,289,925	2,952,169	2,282,005
Chr10	2,698,820	2,714,737	2,642,782	2,649,870	2,413,675	2,564,176	2,483,521
Chr11	2,228,420	2,291,757	2,249799	2,266,931	2,087,508	2,247,110	2,291,537
Chr12	525,065	969,164	857,395	771,183	680,337	978,035	870,680
UC > 500 Kb	-	536,197	-	1,045,802	-	-	-
UC < 500 Kb	257,344	631,872	308,006	473,128	560,738	328,470	339,343

^1^ Chr01-Chr12 corresponds to chromosomes 1–12. Data for all uncharacterized contigs (UC) for each genome were categorised into two, UC > 500 Kbs and UC < 500 Kbs. The total sizes of uncharacterized contigs < 500 Kbs were combined per genome. ^2^ Scaffold sizes are indicated in bp, and the absence of uncharacterized contigs > 500 Kb is indicated with.

**Table 3 jof-08-00657-t003:** Summary statistics for number of reads obtained and mapped to each genome for the different sequencing platforms.

Sequencing Platform	Isolate	Total Number of Quality-Filtered Reads	Number of Reads Mapped to the FSP34 Reference Assembly (%)
Illumina	FSP34	7,840,006	7,762,015 (99.0)
	CMWF560	9,471,099	8,931,113 (94.0)
	CMWF567	9,482,308	9,012,583 (95.1)
	CMWF1803	9,860,143	9,028,628 (91.6)
	UG10	9,737,181	9,394,412 (96.5)
	UG27	9,743,693	8,934,966 (91.7)
	KS17	7,995,046	7,228,461 (90.4)
PacBio	CMWF560	300,889	281,873 (93.7)
	CMWF567	187,327	181,372 (96.8)
	CMWF1803	194,164	183,210 (94.4)
	UG10	256,808	247,772 (96.5))
	UG27	357,305	339,326 (95.0)
MinION	KS17	95,510	92,336 (96.7)
	FSP34	165,477	164,130 (99.2)

**Table 4 jof-08-00657-t004:** Micro variant and structural variant (SV) rate details per chromosome.

Chromosome	Micro Variants ^1^		SVs ^2^			
	Number of Variants	Variant Rate ^3^	Number of Deletions	Number of Duplications	Number of Inversions	Number of Insertions
Chr01	41,809	153	99	1	10	83
Chr02	44,429	114	104	2	14	72
Chr03	35,947	141	97	0	18	69
Chr04	42,594	101	80	1	4	71
Chr05	36,398	121	81	0	8	67
Chr06	44,223	97	112	0	11	68
Chr07	36,258	97	75	3	8	63
Chr08	43,130	73	97	0	4	61
Chr09	31,971	93	60	0	6	44
Chr10	43,870	61	80	0	8	39
Chr11	36,755	60	68	1	10	49
Chr12	17,314	30	33	0	2	31
Total	46,1683	97	986	8	103	717

^1^ Micro variants were determined using Freebayes. ^2^ SVs were determined using Sniffles. ^3^ Variant rate determined by number of variants per base (total chromosome size/total number of variants).

**Table 5 jof-08-00657-t005:** SnpEff annotation summary for macro structural variants (SVs) indicating number of effects. The impact of the different types of effects are indicated in differing colours: High impact (red), moderate impact (orange), low impact (green) and modifier effect (yellow).

Effect Classification	Fields	SnpEff Count	Percentage (%)
Number of effects by impact	High	420,017	88.2
	Low	1	0.0
	Moderate	49,895	10.5
	Modifier	6316	1.3
Number of effects by type	Bidirectional gene fusion	222,121	46.6
	Chromosome number variation	10	0.002
	Conservative in-frame deletion	103	0.02
	Disruptive in-frame deletion	57	0.01
	Downstream gene variant	2499	0.5
	Duplication	1	0.0
	Exon loss variant	100	0.02
	Exon region	2	0.0
	Feature ablation	7724	1.6
	Frame shift variant	149	0.03
	Gene fusion	180,468	37.9
	Intergenic region	608	0.13
	Intragenic variant	367	0.08
	Intron variant	49	0.01
	Inversion	49,863	10.5
	Non-coding transcript variant	310	0.07
	Splice acceptor variant	16	0.003
	Splice donor variant	19	0.004
	Splice region variant	67	0.01
	Splice site region	1	0.0
	Start lost	45	0.01
	Stop gained	1	0.0
	Stop lost	39	0.008
	Transcript ablation	9316	2.0
	Upstream gene variant	2515	0.5
Number of effects by region	Chromosome	49	0.01
	Downstream	2 499	0.53
	Exon	449	0.1
	Gene	438,343	92.1
	Intergenic	608	0.1
	Intron	15	0.003
	Splice site acceptor	5	0.001
	Splice site donor	6	0.001
	Splice site region	1	0.0
	Transcript	31,739	6.7
	Upstream	2515	0.5

**Table 6 jof-08-00657-t006:** SnpEff annotation summary of micro variants indicating number of effects (Freebayes). The impact of the different types of effects are indicated in differing colours: High impact (red), moderate impact (orange), low impact (green) and modifier effect (yellow).

Effect Classification	Fields	SnpEff Count	Percentage (%)
Number of effects by impact	High	18,047	0.7
	Moderate	63,033	2.3
	Low	74,845	2.8
	Modifier	2,563,288	94.3
Number of effects by functional class	Missense	46,372	42.0
	Nonsense	1027	0.9
	Silent	62,991	57.1
Number of effects by type	Conservative in-frame deletion	404	0.02
	Conservative in-frame insertion	396	0.02
	Disruptive in-frame deletion	354	0.01
	Disruptive in-frame insertion	216	0.01
	Downstream gene variant	977,631	35.9
	Frameshift variant	1881	0.07
	Gene fusion	13.881	0.5
	Initiator codon variant	17	0.001
	Intergenic region	277.047	10.2
	Intragenic variant	168.282	6.2
	Intron variant	23.572	0.9
	Missense variant	62.233	2.3
	Non-coding transcript variant	133.146	4.9
	Splice acceptor variant	287	0.01
	Splice donor variant	325	0.01
	Splice region variant	5689	0.2
	Start lost	110	0.004
	Stop gained	1434	0.05
	Stop lost	269	0.01
	Stop retained variant	170	0.006
	Synonymous variant	71,054	2.6
	Upstream gene variant	988,027	36.2
Number of effects by region	Downstream	977,631	36.0
	Exon	136,959	5.0
	Gene	13,881	0.5
	Intergenic	277,047	10.2
	Intron	19,155	0.7
	Splice site acceptor	266	0.01
	Splice site donor	301	0.01
	Splice site region	4518	0.2
	Transcript	301,428	0.0
	Upstream	988,027	36.3

**Table 7 jof-08-00657-t007:** Spine statistics showing the size of the accessory, core and pangenome elements of the *F. circinatum* isolates.

Isolate	Source	Total bp	GC%
FSP34	Accessory	2,669,713	44.19
	Core	42,260,189	47.20
CMWF560	Accessory	3,831,939	45.05
	Core	42,633,795	47.02
CMWF567	Accessory	3,290,975	45.14
	Core	42,465,505	47.06
CMWF1803	Accessory	4,359,907	45.27
	Core	42,501,744	47.22
KS17	Accessory	2,369,668	44.57
	Core	42,008,433	47.42
UG10	Accessory	2,733,552	45.07
	Core	41,905,489	47.72
UG27	Accessory	2,919,797	44.25
	Core	42,449,440	47.06
	Backbone	42,260,189	47.20
	Pangenome	50,076,541	46.85

## Data Availability

The Whole Genome Shotgun project for *Fusarium circinatum* CMWF1803 has been deposited at DDBJ/ENA/GenBank under the accession JAEHFH000000000. The version described in this paper is version JAEHFH010000000. The Whole Genome Shotgun project for *Fusarium circinatum* CMWF560 has been deposited at DDBJ/ENA/GenBank under the accession JAEHFI000000000. The version described in this paper is version JAEHFI010000000. The Whole Genome Shotgun project for *Fusarium circinatum* CMWF567 has been deposited at DDBJ/ENA/GenBank under the accession JADZLS000000000. The version described in this paper is version JADZLS010000000. The Whole Genome Shotgun project for *Fusarium circinatum* UG27 has been deposited at DDBJ/ENA/GenBank under the accession JAELVK000000000. The version described in this paper is version JAELVK010000000. The Whole Genome Shotgun project for *Fusarium circinatum* UG10 has been deposited at DDBJ/ENA/GenBank under the accession JAGJRQ000000000. The version described in this paper is version JAGJRQ010000000.

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
