# Peer review of "Intra-Species Genomic Variation in the Pine Pathogen *Fusarium circinatum"

_jof, 2022, doi:10.3390/jof8070657_

Round 1

Reviewer 1 Report

In the manuscripts of “Intra-species genomic variation in the pine pathogen Fusarium circinatum”, the authors sequenced and assembled five isolates of F. circinatum. They briefly identified some micro and macro structural variants, and discussed the implication of these variants according to SnpEff’s output. They constructed a core genome and depicted the scheme among five isolates. Overall, the manuscripts provide a few informative and valuable for a comparative genome of pine pathogen F. circinatum, which can be a stepping-stone for disease management caused by F. circinatum.  

Major comments:

(1) As there are some published genomes among different isolates of fungi, it’s best to consider the comparative analysis of genomes.

(2) In lines 273-274, chromosome Chr13 can’t be defined so simply. Histochemical stain or other evidence should be provided.

(3) In lines 475-476, these genes were classified as isolate-specific genes, which might be due to the lack of sample. There was no enough evidence to draw the conclusion genes acquired through HGT.

Author Response

Reviewer 1

Comments and Suggestions for Authors

In the manuscripts of “Intra-species genomic variation in the pine pathogen Fusarium circinatum”, the authors sequenced and assembled five isolates of F. circinatum. They briefly identified some micro and macro structural variants, and discussed the implication of these variants according to SnpEff’s output. They constructed a core genome and depicted the scheme among five isolates. Overall, the manuscripts provide a few informative and valuable for a comparative genome of pine pathogen F. circinatum, which can be a stepping-stone for disease management caused by F. circinatum

Major comments:

(1) As there are some published genomes among different isolates of fungi, it’s best to consider the comparative analysis of genomes.

Response: We do not understand the question the reviewer is asking. We request more information to clarify this comment.

(2) In lines 273-274, chromosome Chr13 can’t be defined so simply. Histochemical stain or other evidence should be provided.

Response: The sentence “. We therefore suggest that this scaffold represents an additional chromosome (Chr13) for F. circinatum” has been changed to “The uncharacterized scaffold might be an additional chromosome for F. circinatum.”

(3) In lines 475-476, these genes were classified as isolate-specific genes, which might be due to the lack of sample. There was no enough evidence to draw the conclusion genes acquired through HGT.

Response: The sentence “These were the isolate-specific genes that could have been acquired through HGT” has been changed to “These were mainly the isolate-specific genes based on the sample used in this study.”

Reviewer 2 Report

Fusarium circinatum is an invasive pathogen causing pitch canker on a wide range of pine species in various countries. It is an economically important pathogen. In this study, the authors investigated genomic variation, structural and micro variants, of five geographically diverse isolates using various whole genome sequencing techniques. As the authors claimed, these data would be helpful in developing management strategies for pitch canker disease. Overall, the study is outstanding, and the results are precise. This manuscript is well-written and well-organized. It would be helpful to the readers of the journal.

Author Response

Reviewer 2

Comments and Suggestions for Authors

Fusarium circinatum is an invasive pathogen causing pitch canker on a wide range of pine species in various countries. It is an economically important pathogen. In this study, the authors investigated genomic variation, structural and micro variants, of five geographically diverse isolates using various whole genome sequencing techniques. As the authors claimed, these data would be helpful in developing management strategies for pitch canker disease. Overall, the study is outstanding, and the results are precise. This manuscript is well-written and well-organized. It would be helpful to the readers of the journal.

Response: Thank you for the positive feedback.

Reviewer 3 Report

There are only genome comparisons and anlysis of seven isolates of Fusarium circinatum, lack of biology tests in the paper. So I think the authors should supplement one or two biology tests according to the biological informations.

Author Response

Reviewer 2

Comments and Suggestions for Authors

Fusarium circinatum is an invasive pathogen causing pitch canker on a wide range of pine species in various countries. It is an economically important pathogen. In this study, the authors investigated genomic variation, structural and micro variants, of five geographically diverse isolates using various whole genome sequencing techniques. As the authors claimed, these data would be helpful in developing management strategies for pitch canker disease. Overall, the study is outstanding, and the results are precise. This manuscript is well-written and well-organized. It would be helpful to the readers of the journal.

Response: Thank you for the positive feedback

Round 2

Reviewer 3 Report

The modified version is better than before, and I hope to read your beautiful biology function study of the fungus in future.Thanks a lot.